# Challenge Test as Special Tool to Estimate the Dynamic of *Listeria monocytogenes* and Other Foodborne Pathogens

**DOI:** 10.3390/foods11010032

**Published:** 2021-12-23

**Authors:** Luigi Lanni, Valeria Morena, Adriana Scattareggia Marchese, Gessica Destro, Marcello Ferioli, Paolo Catellani, Valerio Giaccone

**Affiliations:** 1Istituto Zooprofilattico Sperimentale del Lazio e della Toscana “M. Aleandri”, Sede di Roma, Via Appia Nuova 1411, 00178 Rome, Italy; luigi.lanni@izslt.it (L.L.); valeria.morena@izslt.it (V.M.); adriana.scattareggiamarchese-esterno@izslt.it (A.S.M.); 2EPTA NORD Food Analysis & Consulting, 35026 Conselve, Italy; gessica.destro@eptanord.it (G.D.); marcelloferioli@eptanord.it (M.F.); 3Department of Animal Medicine, Productions and Health, School of Agricultural Sciences and Veterinary Medicine, Viale dell’Università 16, 35020 Legnaro, Italy; paolo.catellani@unipd.it

**Keywords:** challenge test, food microbiology, risk assessment

## Abstract

Over 23 million cases of foodborne disease (FBD) occur in Europe each year, with over 4700 deaths. Outbreaks of FBD have a significant impact on our society due to the high economic losses they cause (hospital treatment of affected patients and destruction of contaminated food). Among its health objectives, the European Union has set itself the goal of reducing the incidence of the main FBDs, approving various regulations that codify requirements in order to produce food that is “safe” for human consumption. Among these rules, Regulation 2005/2073 establishes precise food safety criteria for foods that are judged to be most at risk of causing episodes of FBD. The food business operator (FBO) must know their food better and know how to estimate whether a food can support the growth of food pathogens or if they are able to hinder it during the food’s shelf life. It is becoming crucial for each FBO to schedule specific laboratory tests (challenge tests) to establish the growth potential of individual pathogens and their maximum growth rate. In 2008 the European Union published the guidelines for programming the challenge tests for *Listeria monocytogenes* in RTE foods. These guidelines were further implemented in 2014 and again in 2019. In June 2019 the UNI EN ISO 20976-1 was published, which contains indications for setting up and carrying out challenge tests for all foodborne pathogens in all foods. In this article, we compare the three official documents to highlight their common aspects and differences, highlighting the advantages and disadvantages that each of them offers for those who have to set up a challenge test for the various foodborne pathogens. Our conclusion is that the challenge test is today the most effective tool to estimate the dynamics and growth potential of pathogenic microorganisms in food, if it is designed and implemented in a scrupulous way. It is important to develop a rational experimental design for each challenge test, and for each food, and this requires professionals who are experts in this specific field of study and who must be properly trained.

## 1. Introduction

In Europe, 5175 foodborne illness outbreaks were reported in 2019, causing 49,463 clinical cases; 3859 people were hospitalized and 60 people died, 50% more than the deaths recorded in 2018 [1]. The epidemiological data from the European Food Safety Authority (EFSA) confirm that the trend of foodborne disease outbreaks is constantly increasing in Europe. This is happening in spite of the attention that stakeholders have devoted to hygiene practices applied to the production of food for humans as a basic condition to maintain food safety (prerequisite programs). Those basic conditions also include good hygienic practices (GHP) and good manufacturing practices (GMP) as basic conditions and measures, which, together with hazard analysis principle procedures, are necessary to ensure food safety.

The pathogen most often called into question as a source of foodborne illness is *Salmonella*, which caused 926 outbreaks of foodborne illness, 17.9% of all reported episodes. *Salmonella* also caused the highest number of hospital admissions (1915 cases, 49.6% of all hospital admissions) [1]. Furthermore, in 2019 in Europe, there was a significant increase in cases of food listeriosis. *Listeria monocytogenes* (hereinafter referred to as *L. monocytogenes*) caused 349 cases of clinically detectable disease and more than 50% of all fatal cases (31 deaths, compared to 21 deaths registered in 2018 and only two fatal cases registered in 2017) [1]. From 2016 to 2019, the number of cases of food listeriosis registered in Europe has shown continuous growth. As in other member states of the European Union, in Italy in the same period the number of foodborne disease outbreaks increased regularly.

Most outbreaks of foodborne illness are caused by the consumption of foods of animal origin (in particular meat and meat products, but also fishery products, eggs and derivatives, and milk and dairy products). A number of factors contribute to this trend of increasing outbreaks of gastroenteritis and other diseases due to the consumption of contaminated food. In the last 30 years, the lifestyles of Western people have changed and this has drastically influenced the eating habits and methods of preparing meals, especially in the home [2]. In 2019, 25.7 million households were registered in Italy: 33% of these families were made up of a single person, a group comprising 5.9 million people [3]. The increasingly frenetic pace of work and life does not allow most people to have enough free time for home cooking. In addition, when people cook at home, they often have little time to spend on traditional meal preparation. Hence, many people find it convenient to have foods available that require little time for their preparation (ready to cook or pre-cooked) or that are treated in such a way as to reduce the time to prepare them (washing, husking, cutting, etc.), or people opt for ready-to-use ingredients (stuffed pasta, ready sauces, etc.).

It is estimated that today, due to work needs, more than 50% of Europeans are forced to eat at least one meal outside the home during their working day. This has led to a significant increase in the number of points of sale and consumption of food (including the notable spread of street food). In these food delivery points, foods that are already prepared and need only to be cooked (ready-to-cook (RTC)) or that are already precooked or processed are often used, so that they can be consumed without further treatment (ready-to-eat (RTE) products) or only require reheating (ready-to-reheat (RTRH) food).

In retail circuits and in collective catering sectors, RTC and RTE foods are increasingly widespread and are found in increasingly numerous “versions on the theme”, following the increase in requests from consumers who consider them attractive and practical, for the timesaving they allow, although not always at low cost. Unfortunately, RTE and RTRH foods can be contaminated occasionally by foodborne disease microorganisms and, in some cases, the matrix is unable to effectively stop or slow down the multiplication of these microorganisms. If the concentration of the contaminating microorganism exceeds specific threshold values, that food can become concretely harmful to the consumer.

All this led the European Union to issue Commission Regulation (EC) n.2073/2005 of 15 November 2005 “on microbiological criteria for foodstuffs” [4], which was then implemented over time until 2020. Regulation 2073/2005 gave a specific definition to the term “shelf life” for microbiological criteria (well defined in Regulation 1169/2011 [5]) and, in Annex II, attributed a real “legal value” to the challenge test. This has resulted in the issuing of specific rules and guidelines that are used to design and conduct shelf life and challenge tests for microorganisms of interest in food hygiene.

Until the 1990s, challenge testing of food was merely a diagnostic test that could be carried out by choice to assess the degree of sensitivity of a human being to a specific allergen. Only in the early 2000s did we start talking about challenge testing as a specific test to evaluate the ability of a microorganism to develop in a food. The first author who laid the foundations of modern food challenge testing was the Welsh researcher A.D. Russell [6], who published an article in 2003 that outlined the essential concepts underlying the challenge test as we understand it today in the food safety industry.

Since then, various articles have been published documenting the behavior and dynamics of some of the main foodborne disease agents, such as *L. monocytogenes* and *Clostridium botulinum*, due to the intrinsic danger that these bacteria can possess if they manage to contaminate foods and to multiply within them. Among other experimental studies, conducted on various food matrices, we can mention those of Hyytiäa E. et al. [7], Benôıt C. et al. [8], Parker M.D. et al. [9], Varalakshmi S. and Leysen S. [10], Centorotola G. et al. [11], El-Hajjaji S. et al. [12], Gérard A. et al. [13] and Everis L. and Betts G. [14]. However, in the bibliography there are still no articles that have reviewed in a comparative way the guidelines that have governed the design and implementation of challenge testing of pathogenic bacteria in food since 2008. In addition to the European Union, other public [15] or private institutions have already tried to draw up guidelines to regulate the conduct of challenge tests of microorganisms in food. So did, for example, the Chilled Food Association Ltd. In Great Britain in 2010, which published a document entitled “Shelf life of ready to eat food in relation to *Listeria monocytogenes*—Guidance for food business operators” [16].

In 2019, the International Organization for Standardization (ISO) published a specific standard (ISO 20976-1:2019) [17], which now governs the implementation of challenge tests in food worldwide. Following this publication, the European Union supplemented its previous guidelines by issuing a further document in July 2021. On the topic of “challenge testing” of microorganisms in food, therefore, today we have a series of documents on which it is useful to have clear ideas.

The purpose of our article was to compare the guidelines currently existing in Europe and around the world, to obtain information that can be useful for those who have to design and implement challenge tests on food and feed.

In this review, we have compared the methodologies to design and develop challenge tests to calculate the dynamics of *L. monocytogenes* in food because the guidelines currently in force essentially concern this foodborne pathogen. However, remember that the guidelines contained in ISO 20976-1:2019 can be used to plan and develop challenge tests for all genera of microorganisms in all types of food, as well as in animal feed.

## 2. Listeriosis as a Foodborne Disease Caused by Consuming Ready-To-Eat (RTE) Foods: A Brief Overview

Human listeriosis is a serious foodborne disease that can manifest itself with different clinical symptoms, according to the strength of the immune defences of the subject affected. In people with a normally functioning immune system, listeriosis can occur in a completely asymptomatic form or can manifest itself with flu-like symptoms (muscle pain, joint pain, weakness) or as acute gastroenteritis with diarrhoea and fever. In subjects affected by a lowering of immune defences, on the other hand, listeriosis can manifest itself with very serious extra-intestinal symptoms such as hepatitis, pancreatitis, arthritis, and meningoencephalitis, and in the most acute and severe cases, the infection degenerates into septicemia with a high risk of patient death and this leads to high lethality rates of listeriosis in immunocompromised subjects. In pregnant women, listeriosis can cause placentitis and fetal infection, with the possibility of abortion or stillbirth.

Although outbreaks of listeriosis occur rarely, they often cause serious damage and are difficult to resolve. The subjects most at risk are debilitated patients (immunosuppressed subjects, patients with liver diseases, the elderly), in which an invasive form can develop that can end in the death of the affected subject, and pregnant women, with possible abortion, premature births, and birth of infants with neurological lesions.

Reports on foodborne diseases published by public health control bodies worldwide (such as the Centers for Disease Control and Prevention (CDC), Georgia, USA) indicate an increase in the incidence of listeriosis since the 2000s both at the European scale as well as on a global scale. Table 1 shows the numbers of confirmed cases of *L. monocytogenes* in Europe from 2007 to 2020 [18]. In the period between 2007 and 2019, there was an increase of over 1000 cases, as well as a number of hospitalizations and deaths (variable between 8% and 12%). In the year 2020, fewer than 2000 cases were found, but this was most likely linked to the COVID-19 pandemic.

Table 2 shows some of the episodes of foodborne illness that have occurred since the 1980s around the world. According to various authors, the serovars of *Listeria monocytogenes* most involved in clinical episodes have been 1/2a and 4b [19,20,21,22]. In particular, it appears that serovar 1/2a, in the last decade, has been isolated more worldwide than 4b [21,23,24,25,26].

All the foodborne outbreaks reported in Table 2 were linked to the consumption of RTE products based on milk, meat or fish. Among the epidemiological investigations conducted in each of these episodes, it emerged that often the episodes were due to incorrect sanitization of the equipment of the manufacturing companies (delicatessens, dairies, etc.). Another relevant source of contamination could be found at the retail vending stage, when manipulating or slicing the products (slicers, cutting boards, tables, etc.).

## 3. Compliance Criteria for *L. monocytogenes* in RTE Foods: A Global Perspective

Epidemiological data collected in recent years on foodborne disease outbreaks in the most developed countries confirm that listeriosis is a pathology that can have very serious consequences on human health. Its average lethality rate is estimated at around 15–16% both in the European Union [43], as well as in the USA [44]. Nevertheless, in people with low levels of immune defenses the lethality rate of listeriosis can reach over 30–40% of people affected by listeriosis in the clinically evident form.

The severity of listeriosis has led various countries to establish microbiological criteria of compliance for *Listeria monocytogenes* in food. The goal is to reduce the prevalence of listeriosis among the human population gradually.

The Codex Alimentarius Commission (CAC), in turn, published its “Guidelines on the application of general principles of food hygiene to the control of *L. monocytogenes* in foods” in 2007 [45].

In these guidelines, the experts of the Codex Alimentarius Commission established that foods should be classified in two categories as regards the risk from *L. monocytogenes*:

(a) RTE foods in which the growth of *L. monocytogenes* will not occur, and

(b) RTE foods in which the growth of *L. monocytogenes* can occur.

For each of the two categories, the CAC guidelines have established microbiological criteria of compliance with a 2-class plan setting:

(a) for ready-to-eat foods in which growth of *L. monocytogenes* will not occur, the criterion established is that in five sample units *L. monocytogenes* must not exceed 100 cfu/g of food until the end of the commercial shelf life,

(b) instead, for ready-to-eat foods in which growth of *L. monocytogenes* can occur, the criterion the Codex Alimentarius Commission has imposed is given by the absence in 25 g (i.e., <0.04 cfu/g) in five sample units.

The European Union, for its part, had already set its microbiological criteria of compliance for *L. monocytogenes* in RTE foods in 2005. The European Union introduced the compliance criteria for *L. monocytogenes* into law with Regulation n.2073/2005. From that moment on, all FBOs established in the European Union were obliged to comply with these criteria and to set up their own production processes to be able to ensure that each batch of food produced meets the microbiological criteria indicated above.

The Government of Australia has taken up the microbiological criteria for *L. monocytogenes* in RTE foods suggested by the Codex Alimentarius Commission in its guidelines. The new criteria for *L. monocytogenes* in ready-to-eat (RTE) foods were officially published in the Food Standards Code on 31 July 2014 [46]. For Australian producers of RTE food, the criteria for *L. monocytogenes* established are as follows:

(1) less than 100 cfu/g at the end of shelf life for RTE foods in which the growth of *L. monocytogenes* will not occur,

(2) *L. monocytogenes* not detected in 25 g of RTE foods in which the growth of *L. monocytogenes* will not occur.

Canada updated its microbiological criteria of compliance for *L. monocytogenes* in RTE food in 2011 [47]. According to the Canadian regulations currently in force, RTE food is divided into two categories:

(a) category 1 contains products in which the growth of *L. monocytogenes* can occur; and

(b) category 2 contains two subgroups:(2A) RTE food products in which limited growth of *L. monocytogenes* to levels not greater than 100 cfu/g can occur throughout the stated shelf life and(2B) RTE food products in which the growth of *L. monocytogenes* cannot occur throughout the expected shelf life of that food.

The compliance criteria for L. monocytogenes established by law by Canada are as follows:for the RTE foods in which the growth of *L. monocytogenes* can occur throughout the stated shelf life: absence of *L. monocytogenes* in 5 samples units × 25 g of analytical unit,for category 2A (i.e., RTE foods in which a limited potential for growth of *L. monocytogenes* to levels not greater than 100 cfu/g can occur throughout the stated shelf life), the concentration of *L. monocytogenes* must remain <100 cfu/g in 5 samples units × 10 g of analytical unit,for the category 2B (i.e., RTE foods in which growth of *L. monocytogenes* cannot occur (<0.5 log cfu/g increase throughout the stated shelf life), the concentration of *L. monocytogenes* must remain <100 cfu/g in 5 samples units × 10 g of analytical unit.

The US government health authorities, on the other hand, are much stricter regarding the microbiological compliance criteria for *L. monocytogenes* in RTE foods [47]. In fact, the U.S. Department of Agriculture’s Food Safety and Inspection Service (FSIS) maintains a zero-tolerance policy for *L. monocytogenes* in ready-to-eat meat and poultry products [48]. In practice, the US government authorities require that in any type of RTE foods, with no distinctions, the microbiological criterion to be adopted is the absence of *L. monocytogenes* in 5 sample units of 25 g each. In case of a positivity to the research of *L. monocytogenes*, the recommended Action (as literally written in the FSIS report), if limit is exceeded, is “Reject lot. Investigate cause of contamination. Determine if other lots are involved. Determine steps to prevent reoccurrence” improving the adoption of corrective actions.

Comparing with each other the various compliance criteria for *L. monocytogenes* proposed by various countries, we could note that the general rule followed by all governmental authorities is always quite the same. An RTE food is classified as able to allow the growth of *L. monocytogenes* if its growth potential during the commercial life of the food is >0.5 log. Conversely, an RTE food is classified as not able to permit the growth of *L. monocytogenes* if the bacterium has a growth potential of <0.5 log in that food.

In addition to evaluating the pH and/or A_w_ characteristics of the food, therefore, it becomes very important to be able to calculate the growth potential of *L. monocytogenes* in a food. The challenge tests that are the subject of our review are aimed at this objective.

## 4. The EC Regulation n.2073/2005

The EC Regulation n.2073/2005 is intended for food sector operators. According to this Regulation, anyone who produces food in one of the Member States or exports to the European Union is obliged to implement the internal own-checking procedures that allow them to keep under control the hygiene of their production processes and the finished products.

The obligation derives from the fact that the EC Regulation 2073 establishes precise criteria of conformity (criteria of food safety as well as criteria of process hygiene) for some categories of foods which, based on epidemiological data, are considered to be the most at risk of causing episodes of foodborne illness in humans.

To comply with the criteria established by the Regulation 2073, each Food Business Operator (FBO) is obliged to evaluate the effectiveness of its systems in all stages of production, processing and distribution of the foods subject to their control. The FBO is also required to write, implement and maintain permanent procedures (according to the seven main principles of the HACCP system) that result in preventive measures and/or corrective actions, to avoid non-compliance such as failure to comply with a microbiological criterion.

The main purpose of Regulation 2073/2005 is to oblige the FBO to implement and demonstrate that the proposed system is able to keep the hygiene of its production processes under control.

The microbiological criteria established by EC Regulation 2073 are of two types:

Food safety criteria that apply to specific categories of foods when they are placed on the market (not before) and that apply to the entire shelf life of the product, and

Process hygiene criteria that apply to the food processing process and that apply during the food processing stages, but not in the food product when it is on the market and in distribution.

The criteria set by EC Regulation 2073 are criteria defining the acceptability of a product, a batch of foodstuffs or a process, based on the absence, presence or number of microorganisms, and/or on the quantity of their toxins/metabolites, per unit(s) of mass, volume, area or batch. In particular, food safety criteria are parameters defining the acceptability of a product or a batch of foodstuff applicable to products placed on the market.

According to the Regulation 2073, however, the process hygiene criteria are criteria indicating the acceptable functioning of the production process. Such criteria are not applicable to products placed on the market. They set an indicative contamination value above which corrective actions are required in order to maintain the hygiene of the process in compliance with food law.

The rules contained in EC Regulation 2073 apply to many of the food industries; consequently, each FBO is required to comply with them. Each FBO must therefore ask himself whether with that type of production process he is able to comply with the criteria of process hygiene and food safety that is/are imposed on him by Community legislation [49,50]. 

If with analytical checks it emerges that the required criteria are met, this means that:the FBO has carried out a correct design of the documentation,the application and verification of the HACCP self-control plan are valid;the production process is under control;the batches of food produced are (at least in theory) all compliant with the microbiological criteria provided.

If, on the contrary, the results of the analyses show that the microbiological criteria established by the Regulation 2073 cannot be respected, this means that the HACCP plan does not provide sufficient guarantees to comply with the general obligations (see Article 3) and specific (see Article 4) of EC Regulation 852/2004 [51]. In this case, the FBOs shall review or implement the permanent procedure according to article 5 of that Regulation.

To be exact, Article 3 of Regulation 852/2004 expressly provides that: “Food business operators shall ensure that all stages of production, processing and distribution of food under their control satisfy the relevant hygiene requirements laid down in this Regulation.”. Instead, the article 4 point 3 of the same Regulation expressly provides that: “Food business operators shall, as appropriate, adopt the following specific hygiene measures: (a) compliance with microbiological criteria for foodstuffs;”. 

With this article, Regulation 852/2004 announces the establishment of specific microbiological criteria for food intended for human consumption, criteria which will then be made official with the publication of EC Regulation 2073/2005 we are dealing with.

In particular, the drafters of EC Regulation 2073 have reserved specific attention to ready-to-eat foods, which have been defined by the same Regulation as follows: “ready-to-eat food means food intended by the producer or the manufacturer for direct human consumption without the need for cooking or other processing effective to eliminate or reduce, to an acceptable level, the microorganisms of concern”.

For these RTE foods, the EC Regulation 2073 has provided, for *L. monocytogenes*, different food safety criteria, precisely due to the role that this bacterium has as a foodborne pathogen.

The FBO that produces RTE foods therefore has the opportunity to comply with one of two possible food safety criteria for *L. monocytogenes*, based on the category attributed and verified by the FBO to its RTE food.

The FBO must evaluate the physico-chemical characteristics of its product (in particular the pH and A_w_ values of the food to determine if it should be included in one of the two categories provided for by the EC Regulation 2073, classifying its ready-to-eat food among those are 

able to support the growth of *L. monocytogenes*, or alternativelyare unable to support the growth of *L. monocytogenes*.

The food safety criteria for *L. monocytogenes* vary according to this classification. In the first instance, therefore, the FBO can classify an RTE food as a matrix able or unable to support the growth of *L. monocytogenes* based on the pH and/or A_w_ values that the product has and that the FBO periodically records in the self-check phase.

In fact, according to the Regulation 2073: “Products with pH ≤ 4.4 or A_w_ ≤ 0.92, products with pH ≤ 5.0 and A_w_ ≤ 0.94, products with a shelf life of less than five days shall be automatically considered to belong to this category. Other categories of products can also belong to this category, subject to scientific justification.”

According to the EC Regulation 2073, the FBO can classify a RTE food in the category of those unable to support the growth of *L. monocytogenes* regardless of the pH and/or A_w_ values that it has, if it can demonstrate, with well-founded scientific justification, that that food is effectively unable to support the growth of *L. monocytogenes*.

The challenge test is (today) the most comprehensive and scientifically based approach that we can have to establish whether an RTE food is able or unable to support the growth of *L. monocytogenes*.

We can always deduce it from the Annex II of the EC Regulation 2073 where verbatim we find written: 

“When necessary on the basis of the abovementioned studies, the food business operator shall conduct additional studies, which may include:predictive mathematical modeling established for the food in question, using critical growth or survival factors for the microorganisms of concern in the product,tests to investigate the ability of the appropriately inoculated microorganism of concern to grow or survive in the product under different reasonably foreseeable storage conditions,studies to evaluate the growth or survival of the microorganisms of concern that may be present in the product during the shelf-life under reasonably foreseeable conditions of distribution, storage and use”.

Annex II itself of the Regulation 2073 ends with this sentence: “The above-mentioned studies shall take into account the inherent variability linked to the product, the microorganisms in question and the processing and storage conditions”.

In conclusion, Annex II suggest to the FBO to consider also other intrinsic characteristics of the matrix (moisture content, Redox potential, nutrient, presence of antimicrobial additives, biological structures) as well as the extrinsic environmental conditions (temperature of storage, relative humidity of environment, presence and concentration of gases, presence and activities of other microorganisms. These sentences of the EC Regulation 2073 contain all the fundamentals of what must be satisfied as an approach to design and to perform a well-founded challenge test. The voluntary reference documents that are a topic of comparison in our article reproduce exactly the indications of the community legislators who wrote the EC 2073 Regulation.

## 5. Guidelines for Shelf Life Studies of *L. monocytogenes*: A Brief History

The first national institution to draw up guidelines for designing and conducting challenge test studies for *L. monocytogenes* was the AFSSA (Agence française de sécurité sanitaire des aliments), an independent state office which in France was responsible for detecting the appearance of possible health and hygiene risks in food for humans and animal feed, and to study their evolution. The experts of the French AFSSA developed and published the first guidelines for conducting challenge tests for *L. monocytogenes* in food [52].

On 1 July 2010, the AFSSA was abolished and the ANSES (Agence nationale de sécurité sanitaire de l’alimentation, de l’environnement et du travail) was created in its place. The ANSES has inherited all the functions of control body carried out in previous years by the AFSSA, further extending them in other fields besides that of food safety. This explains why the first document we considered were the AFSSA guidelines of 2008. This first document was developed by the French AFSSA experts in collaboration with experts from seven other laboratories, including six national reference laboratories for *L. monocytogenes*.

The AFSSA guidelines 2008 were reviewed by Beaufort [53], but it was still very complex to apply, not exactly user-friendly. For this reason, in 2012 the ANSES decided to form a working group made up of representatives of ten national reference laboratories.

The joint work of this working group led to the publication of a second version of the guidelines for conducting challenge tests for *L. monocytogenes*, published in 2014 under ANSES standards as a community reference laboratory for *L. monocytogenes*. This is the second document we have considered.

In February 2019, the ANSES also published an amendment to its 2014 guidelines, in which only minor adjustments were made to the general layout of the pre-existing guidelines. The most significant update made by the 2019 amendment was the remodeling of the conditions of thermal abuse to which the foods must be subjected during the challenge test. In details, the last part of the thermal abuse tests can be conducted at 10 °C and not at 12 °C as provided for in the original text of the ANSES 2014 guidelines. The amendment to the ANSES guidelines of 2019 is the third document we have compared.

Finally, on 17 April 2019 the International Organization for Standardization (ISO) published its specific standard, the EN ISO 20976-1: 2019 “*Microbiology of the food chain—Requirements and guidelines for conducting challenge tests of food and feed products—Part 1: Challenge tests to study growth potential, lag time and maximum growth rate*”. 

This ISO standard is currently the reference text for the design and implementation of all the challenge tests to study growth potential, lag time and maximum growth rate in raw materials and intermediate or finished products. This document is the fourth that we have considered for our comparison.

On July the 1st 2021 the working group formed at the time by ANSES published a further document which is intended to complete the indications provided by the ISO 20976-1 standard when it comes to designing and conducting challenge tests for *L. monocytogenes* in food. The “*EURL Lm Technical guidance document on challenge tests and durability studies for assessing shelf-life of ready-to-eat foods related to L. monocytogenes*” (version 4 July 2021) is the fifth document that we have taken into consideration for our comparison.

## 6. Methodology for the Development of Comparative Evaluation

To conduct our comparative assessment, we took into consideration the English texts of the following documents:EN SANCO/1628/2008 ver. 9.3 [54].AFSSA “Technical guidance document on shelf life studies for *L. monocytogenes* in ready-to-eat-foods, CRL for *L. monocytogenes*”, version 2—November 2008,ANSES “EURL Lm technical guidance document for conducting shelf life studies on *L. monocytogenes* in ready-to-eat foods”, Version 3—6 June 2014 [17]ANSES “EURL Lm technical guidance document for conducting shelf life studies on *L. monocytogenes* in ready-to-eat foods”, Version 3 of 6 June 2014—Amendment 1 of 21 February 2019 [17]ISO 20976-1 First edition 2019-03 Microbiology of the food chain—Guidelines for conducting challenge tests of food and feed products—Part 1: Challenge tests to study the growth potential, lag time and the maximum growth rate”ANSES “EURL Lm Technical guidance document on challenge tests and durability studies for assessing shelf-life of ready-to-eat foods related to *L. monocytogenes*, version 4 of 1 July 2021”.

The aim we set ourselves was not to simply review the similarities and differences that exist between the various documents mentioned above. Our aim has been, rather, to compare the basic concepts of the various documents examined, in order to:
(a)highlight the evolution of the basic principles on which the overall concept of challenge test is based, seen as the main analytical tool to “measure” the growth potential of microorganisms harbored in a food, when they are tested with the components of food matrix itself,(b)better frame the most relevant aspects of the challenge test, overlapping the concepts contained in each single document consulted,(c)highlighting any detailed elements among the documents useful as a complete integration for the design of experiment.


In fact, our conclusion is that to design a scientifically well-founded challenge test and therefore to obtain results that are effectively predictive, it is not enough to study and implement what is contained in the ISO 20876-1 standard alone. We need also to associate the guidelines provided by the ANSES EURL Lm 2021, at least as regards the challenge tests to be carried out to determine the growth potential and/or the maximum growth rate of *L. monocytogenes* in food.

## 7. Comparative Critical Survey

To develop this part of the article, we examined the individual elements (components or phases) of a challenge test, comparing the concepts, definitions and/or indications that emerge from their comparative comparison. The evaluations and comments that we will present by points are a reasoned synthesis of our reflections, as they emerged from the comparison. We have divided our assessments into individual topics and within each topic we will report our specific considerations.

### 7.1. Useful Terms and Definitions

#### 7.1.1. Thermal Abuse

The first document that took into consideration this specific storage condition of the food units subject to challenge test is the AFFSA 2008 text which identified it as follows: “temperature higher than the prescribed temperature of processing and retail storage according to national legislated temperature rules or European food regulations, including reasonably foreseeable domestic storage conditions. The abuse temperature covers the whole cold chain, taking into account in particular the temperature deviation of retail refrigerators as well as domestic storage”.

Annex E of ISO 20976-1 also takes into consideration the storage conditions specifying that “FBOs shall ensure that the food is microbiologically acceptable, throughout its shelf-life, under reasonably foreseeable storage conditions at the manufacturer, retailer and consumer levels, including some possible time and temperature abuses. The storage conditions and particularly the temperature can significantly influence microbial population. As previously mentioned, storage conditions (time and temperature) data can be based on observations from the different steps and the countries where the food supply chain and consumers are located. It is also possible to use:data regarding temperature monitoring along the food chain available in the scientific literature;practical monitoring of storage temperatures in the studied cold chains.

A challenge test provides the estimated growth rate at a fixed temperature. This can be used for further simulations in both static (e.g., a different pH than the pH of the tested food matrix) and dynamic conditions (e.g., temperature change or fluctuations along the food chain) using predictive modeling based on the gamma-concept”.

#### 7.1.2. Batch

There are two documents that defined it, the AFFSA 2008 guidelines and the ISO 20976-1: 2019 standard. In both cases the definition is the same, i.e., “Group or set of identifiable food obtained through a given process under practically identical circumstances and produced in a given place within one defined production period”.

#### 7.1.3. Challenge Test

The only formally reported definition is that present in the ISO 20976-1 standard for which the challenge test (point 3.5) is “study of the growth or inactivation of microorganism (s) artificially inoculated in a food”.

In EURL Lm 2021, however, we find an informal definition of challenge test, identified as a “laboratory-based study used to evaluate the microbiological safety of a product”. According to the EURL Lm 2021, it is significant that the challenge test is not seen only as a simple analytical test to determine a result (the growth potential of *L. monocytogenes* in a given food), but also and above all as a diagnostic tool that allows the FBO to assess the microbiological safety of its product with better predictivity.

#### 7.1.4. Control Units

They are precisely defined only in ISO 20976-1:2019 and, consequently, the ANSES EURL Lm 2021 only makes a reference to the standard ISO. The two documents insert an essential requirement in the design of a challenge test: in addition to the food units that are inoculated with the bacterium to be tested, in the general scheme of the challenge test it is also necessary to provide additional food packages, identical to those inoculated. These additional test units must be inoculated with a volume of sterile physiological solution equal to that of the bacterial suspension, to evaluate any influences due to a change in the actual composition of the food which, in practice, results in the same in "same chemical-physics conditions of inoculated test units". These additional test units, which are associated with those inoculated with the test bacterium, must be kept under the same conditions as the test units inoculated with the test bacterium and will be used to determine the pH, A_w_, and other factors. intrinsic and extrinsic to the product, as well as the “background” microbial flora harbored in the food.

#### 7.1.5. Food Control Sample

It is a food unit different from the control unit, it is subjected to any preparation and is used to verify the representativeness of the production.

#### 7.1.6. Growth Potential

The only definition that we find in the documents compared with each other is that reported in the ISO 20976-1 footprint and compared to the indications contained in the previous ANSES guidelines it constitutes one of the most important innovations brought by ISO 20976-1.

The growth potential is a number that expresses, in logarithmic values, how much a bacterium or yeast can grow in that particular food, over the shelf life of the latter and in the programmed storage conditions (refrigeration and thermal abuse).

Specifically, the growth potential of the microorganism subject to challenge test is obtained by making the difference between the highest concentration of the bacterium recorded during the challenge test (indicated as log_max_) and the initial concentration of the inoculated bacterium (log_i_).

In the previous ANSES guidelines, however, the growth potential was obtained by subtracting the logarithm of the initial concentration inoculated from the logarithm of the concentration of the same bacterium recorded at the end of the experimental period (t_end_). The conceptual difference that emerges from the ISO 20976-1 standard compared to the previous EURL guidelines is remarkable: in fact, the ISO records the real development dynamics of a bacterium (for example *L. monocytogenes*) more faithfully than the EURL Lm 2008 and 2014 temporarily increase in concentration during the shelf life of the product, and then decreases again in the final part of the shelf life of the food.

In this respect, the ISO 20976-1: 2019 is more realistic and adherent than the EURL Lm to the dynamics that microorganisms can have in a food, during its commercial life.

#### 7.1.7. Organizing Laboratory

The ISO 20976-1 standard introduces this definition which is not present in the ANSES EURL Lm documents. According to the ISO 20976-standard, the “Organizing laboratory” is the laboratory that has the responsibility of “managing” the challenge test, that is, putting into practice the project design of the challenge itself. In our opinion, it is implicit in the definition that the laboratory organizing the challenge test is an accredited laboratory to carry out analyses in the agro-food sector, but strictly speaking ISO 20976-1 is not so explicit. Indeed a reference to this is reported in point 4.1 The analyses shall be conducted under a quality assurance system (e.g., in accordance with ISO/IEC 17025).

#### 7.1.8. Sampling

The concept is introduced in the challenge test design by ISO 20976-1 as a “Selection of one or more units or portions of food such that the units or portions selected are representative of that food”. It is important to keep in mind that the test units used in the challenge test must be representative of the food that is the subject of the test.

#### 7.1.9. Scope

This is one of the most important topics in the overall challenge test scenario. According to the ANSES EURL Lm 2014 document, the scope of the challenge test “is to simulate as closely as possible the likely storage conditions of the products”. ISO 20976-1 is much more rational and expressly specifies that “This document specifies protocols for conducting microbiological challenge tests for growth studies on vegetative and spore-forming bacteria in raw materials and intermediate or end products”.

It is evident that according to the ISO rules it is now possible to design and carry out a challenge test on any type of food, whether it consists of raw materials, intermediate products, or finished products for numerous microorganisms of food interest. In this respect, ISO 20976-1 exceeds the previous ANSES EURL Lm guidelines, which limit challenge testing to finished products only and moreover to products in small mass test units, intended for the final consumer. The ANSES EURL Lm 2021 openly declares that its guidelines must be associated with the rules contained in ISO 20976-1 regarding only the challenge tests conducted to calculate the growth potential of *L. monocytogenes*.

#### 7.1.10. How Many Batches to Examine of the Same Product?

As noted in the ISO 20976-1 standard, the number of food batches to be subjected to the challenge test depends on the degree of variability of the production process, which is reflected in the chemical-physical characteristics of the food. In particular, the inter-batch variability can be determined on the basis of the pH and A_w_ values of the product, but also on the basis of other factors intrinsic or extrinsic to the food, such as the addition of preservative additives or the development in the product of an abundant population of lactic bacteria which with their metabolism can counteract the development of other bacteria, including the bacteria most often subject to challenge tests, which are the main agents of foodborne illness.

The ANSES EURL Lm 2014 already stipulated that individual batches of food should be sent to the laboratory as soon as possible after production and (in a footnote) suggested limiting the time that could elapse between food production to 48 h. to be tested and the inoculation of the test units from that lot.

This “rule”, expressed almost in passing in the ANSES EURL Lm 2014, was then taken up in a much clearer form by the ANSES EURL Lm 2021. The ISO 20976-1: 2019 standard does not refer in point 7.2 (Number of batches and selection criteria) to the time that must elapse between production and inoculation, but this concept is well highlighted in point 13 (Analysis) where it is written, “For all types of challenge tests, an initial analysis shall be performed on the day the food is inoculated and at all subsequent sampling points relevant to the type of challenge test being performed”.

The general rule, however, is to challenge at least three batches of the same food, as highlighted in point 7.2 of ISO 20976-1 “Otherwise, a minimum number of three batches should be used for both growth potential and the growth kinetics studies. If there is a significant variability of the rheological as well as chemical characteristics of the product between the production batches, especially as regards the pH and A_w_ values, and you want to study the growth rate, more than 3 batches should be analyzed”.

The ISO 20976-1 standard therefore better specifies what the previous ANSES guidelines had already more or less clearly suggested, namely, that the challenge test can be limited to a single batch of food as long as there are clear and obvious reasons, for example, citing the ISO 20976-1 standard “(a) evaluating the impact of a new formulation of the food; (b) using a batch representing the most favorable growth conditions (worst case); (c) applicable for a growth kinetics study only if the impact of the inter-batch variability determined by the calculator tool (see Annex A) is not significant”.

#### 7.1.11. Choice of Strains to Inoculate

All the documents that we have reviewed in a comparative way agree that a challenge test is correctly set up if at least two or three different strains of the same microorganism subject to the challenge test are inoculated in the food.

Strains isolated from the food matrix or from the production environment are preferred compared to strains from culture type collections and those isolated from the same food matrices or from the production environments subject to the challenge test being designed are preferred.

The purpose of the ANSES EURL Lm guidelines and of the ISO 20976-1 standard itself, which follows the same concept, is evident: by mixing together two or three different strains of the same microorganism (of which one or two are of “wild” origin) it is possible to better consider the variations in behavior (growth/survival) that certainly exist between single strains of the same microorganism.

The 2014 ANSES guidelines did not require precise knowledge of the wild strains used in the challenge test. The ISO 20976-1 standard, on the other hand, and the ANSES EURL Lm 2021 take a decisive step forward, which is very significant from this point of view. In fact, both the ISO 20976-1 standard and the ANSES 2021 explicitly require that each strain used in the challenge test must be “well characterized” from a biochemical and/or serological and/or genomic point of view, at least “in sufficient detail for its identity to be known“ (ISO 20976-1), as well as in terms of its growth ability. The ANSES EURL Lm 2021 also specifies that the European Union reference laboratory network for *L. monocytogenes* has the task of collecting and typing a series of wild strains that are sent to them from the peripheral laboratories and thus creating a “bank of well-typed strains” which, in turn, can be sent on request to laboratories that intend to conduct challenge tests on human food and/or feed. The rule of using two or three strains of the microorganism subject to challenge testing in a mixture is valid if the test is aimed at determining the growth potential of the inoculated microorganism. If, on the other hand, the challenge test is designed and conducted to estimate the growth rate of the microorganism, only one strain of that microorganism will have to be used (logically).

#### 7.1.12. Inoculation of Test Units

This is a particularly important part of the challenge test because it involves carefully programming the ways in which the microorganism subject to the challenge test comes into contact with the food to be evaluated. Here the ISO 20976-1 standard and the subsidiary ANSES EURL Lm 2021 differ slightly from each other in dictating the rules:

(1) The ISO 20976-1 standard provides that the challenge test can be conducted on any type of food (and animal feed). These can be small portions of food, in original packaging already suitable for retail sale, as well as intermediate products or even raw materials, even starting from relevant food masses (e.g., an entire cheese wheel or a whole cured raw ham). ISO 20976-1 merely provides that in the case of food in small packages, the microbial suspension is inoculated in the entire mass of the food. If, on the other hand, it is a question of food samples of a large mass, the representative test units must be taken from the original sample, obviously strictly respecting the rules of asepsis (aseptically).

(2) ANSES EURL Lm 2021 for *L. monocytogenes* alone, on the other hand, focuses its attention above all on the methods of inoculation of the test units. These guidelines in fact recommend ensuring a homogeneous dispersion of the microbial suspension in all parts of the food, “even if in reality this may not be the case”.

The ISO 20976-1 standard and, consequently, the ANSES EURL Lm 2021 guidelines also provide that the test units subject to inoculation are kept in the identical packaging conditions typical of that product. This means that for all foods it will be necessary to keep the original packaging or in any case use packaging material of the same type in the event that the original food packages have to be opened to carry out the inoculation of the individual test units.

Logically, in the case of foods packaged in MAP, it is necessary to ensure that the inoculation of the microbial suspension does not substantially change the composition of the protective atmosphere used by the manufacturer. In the case of foods originally packaged under vacuum, however, it is inevitable that the packages will be opened to carry out the inoculation. In this case, the laboratory technicians will have to ensure that, after inoculation, the test units are packaged again with the same vacuum specifications.

The ISO 20976-1 standard wisely recommends inoculating a number of “additional” test units with respect to those actually needed to conduct the challenge test; in this way it will be possible to replace any test units that were damaged during the challenge test.

All the strains present in the inoculum cocktail must be at the same concentration in all the documents compared by us, finally; the microbial suspension that is inoculated in the test units must not exceed 1% of the mass or volume of the unit of test itself. This, logically, is aimed at avoiding excessive modification of the physicochemical characteristics of the food, in particular its pH and especially its A_w_.

#### 7.1.13. Storage Conditions of the Test Units

Each challenge test is made up of phases (or “topical moments” if we want to call them that) and each of them is of considerable importance for the correct design of the entire test and for achieving scientifically acceptable results. This concept also applies to the conditions under which test units must be kept for the duration of the challenge. This is confirmed by the fact that the correct management of this phase had already been taken into consideration by ANSES EURL Lm 2014, which limited itself, however, to prescribing that the test units had to be maintained according to "the conditions at which the product is most likely to be subjected in normal use, until his final consumption". To better specify their intentions, the drafters of ANSES EURL Lm 2014 even prepared a specific table, contained in the guidelines, precisely indicating the conditions under which the test units had to be maintained during the challenge test.

The ISO 20976-1 standard deviates significantly from the very “pragmatic” approach of the ANSES guidelines. The ISO, in fact, limits itself to establishing that “All test units shall be stored (5.2. and 5.3.) At the selected temperature(s) for the appropriate time. The temperature throughout the duration of the challenge test shall be recorded. The integrity of the packaging shall be maintained throughout the duration of the challenge test. The temperature chosen for storage of the test units should allow growth of the target microorganism and be as close as possible to the reasonably foreseeable food storage conditions. For food affected by natural environment conditions (e.g., relative humidity), it may be necessary to use a climate-control chamber (5.3.) That can mimic those storage conditions. In this case, the relevant environment conditions inside the chamber shall be recorded throughout the duration of the challenge test.”

In summary, therefore, the ISO 20976-1 standard leaves room for the FBO who carries out the design of their challenge test to determine the storage methods for the test units and limits itself to “suggesting” that to plan and then carry out this phase of the challenge, FBO should replicate as faithfully as possible the “reasonably foreseeable” storage conditions, which the FBO is called upon to choose.

In the case of challenge tests for *L. monocytogenes*, however, it must be borne in mind that the ANSES EURL Lm 2021 guidelines continue to provide, with the usual pragmatism, that the test units must be kept at 7 °C for the first two phases of the food chain (manufacturer and retail levels) and at 10 °C for home preservation. The only significant innovation, from this point of view, is that the temperature at which to store the test units at home has dropped from 12 °C to 10 °C, mitigating the extreme conditions of thermal abuse that were expected from the lines. ANSES EURL Lm 2014 guide.

#### 7.1.14. Calculation of Growth Potential

In this “topical moment” we find perhaps the point of greatest differentiation between the previous ANSENS EURL guidelines and the ISO 20976-1 standard (of which the concept is fully accepted by the latest ANSES EURL Lm of July 2021 for the challenge tests of *L. monocytogenes*).

According to the ANSES EURL Lm guidelines which were in force before the entry into force of the ISO 20976-1, the growth potential of *L. monocytogenes* (and theoretically that of all other bacteria that could possibly be used in a challenge test) derived from the difference between the concentration of *Listeria* at the expiry of the shelf life of the product and the initial concentration of the bacterium at the time of its inoculation.

The ISO 20976-1 standard, on the other hand, establishes that the growth potential (also expressed as a Δ-value) must consider all the programmed experimental data points; in fact, it is first necessary to identify the one where the maximum concentration value is reached, from which the initial inoculated concentration value must be subtracted. In this way it is possible to evaluate the fluctuation of the concentration during the whole period of time considered.

In summary, for the ISO 20976-1 standard Δ = log_max_ − log_i_, where log_max_ is the highest concentration reached during the challenge by the inoculated microorganism and log_i_ is its initial concentration.

The ISO 20976-1 standard does not foresee negative growth potentials: if, throughout the duration of the challenge test, the highest concentration of the inoculated microorganism remains the initial one, the growth potential will be calculated as zero (0).

The final delta in the case of the challenge tests conducted on three product batches will logically be the highest growth potential value recorded between the individual batches. It is on the growth potential value that is based the decision of the FBO to classify its product in the category of foods “which allow” or, vice versa, “which do not allow” the development of the microorganisms tested (and in particular of *L. monocytogenes*).

From this point of view, we note a partial differentiation between the ISO 20976-1 standard and the ANSES guidelines; for the latter the rule still applies that:
(1)If the growth potential value is ≤0.5 log, the food is judged not able to allow the growth of *L. monocytogenes*,(2)If the growth potential value is >0.5 log, the food is judged to be able to allow the growth of *L. monocytogenes*.


The ISO 20976-1 standard, on the other hand, on this specific aspect (which is very important) adopts a “more nuanced position”: in principle, the ISO follows the ANSES guidelines by providing as a discriminating parameter the punctual value of 0.5 log (≤0.5 or >0.5 log). However, the ISO 20976-1 standard assumes that on certain occasions it is possible to accept, as a discriminating value, even a growth potential of 1.0 log without, however, providing more precise indications in this regard.

#### 7.1.15. Exploitation of the Results

This is a phase of the challenge test that is generally never valued sufficiently, although in our opinion it is a particularly important component of the work. From this point of view, the ISO 20976-1 standard does not enter into the merits and does not provide indications. Instead, the ANSES EURL Lm guidelines do so, both in the 2014 version and in the most recent version from 2021. According to these documents, the FBO is responsible for the use of the results of the challenge test. 

It is the task of the laboratory to conduct the challenge test in the manner described above, but once the final result (the growth potential value) is reached, it is the responsibility of the FBO to decide whether to consider its food capable of allowing specific microbial growth or not, based on the shelf life attributed to that food. It could not be otherwise—it is the FBO who knows the characteristics of his product thoroughly and who manages the production process, including the periodic checks carried out according to the HACCP plan that he himself activates in his plant.

Another decision that can be very important is also up to the FBO, namely, assigning the correct value to the estimated growth potential value, in view of the shelf life of its product. In other words, today it is essential for the FBO to take into account the growth potential value of the microbial populations that can degrade the food and of the pathogenic bacteria to establish a correct shelf life. If the challenge test shows that the shelf life of a food (estimated with a storage test) is excessive compared to the growth potential value, the FBO must have the sensibility to admit that its food could become unsuitable and/or harmful for the consumer, if that shelf life were maintained. It is the responsibility of the FBO, in these cases, to decide to shorten the shelf life of one of its products, bringing it to a duration that will not allow the food to reach concentrations that could be critical or potentially dangerous for the health of consumers.

#### 7.1.16. Test Report

All the phases in which a challenge test is articulated must be documented in a final report. The documents we compared do not explicitly indicate who should write this report. However, especially based on the provisions of the ISO 20976-1 standard, it is clear that it is up to the analysis laboratory to write the final report, which is then delivered to the FBO. In fact, the ISO 20976-1 standard states, “The test report shall state the method used and the results obtained with the standard errors mentioned in 14.2 and 14.3. It shall also give details of all operational steps that are either not specified or regarded as optional in this document, and shall report any deviations that might have influenced the results. The test report shall include all the data needed for interpreting the challenge test.”

ISO 20976-1:2019 also indirectly provides the chapters, paragraphs, and details that must be described in a report (e.g., chapter: (i) Aim of the study and type of challenge test (15.2); (ii) paragraph: characteristics of the food relevant to the test; (iii) details: food composition and structure when known including all ingredients, food additives and photograph to illustrate food composition, structure and packaging).

The mandatory nature of the final report of the entire challenge test was already foreseen in the ANSES EURL Lm 2014 guidelines. The ANSES EURL Lm 2021, refers directly to the ISO 20976-1:2019 standard very simply with the phrase “for all the points to include in the test report”.

## 8. Results Obtained from the Comparison

The comparative evaluation that we conducted on the technical documents AFSSA and ANSES and on the ISO 20976-1 standard dedicated to the design and execution of challenge tests for microorganisms of food interest, has allowed us to derive a series of indications, which were reported below and the key points are reported in Table 3.

(a) The challenge test is an experimental test that can be aimed at determining two distinct parameters of the evolutionary capacity of any bacterium or yeast, if inoculated into a food: (i) the growth potential over the shelf life of the product and (ii) the maximum growth rate;

(b) The challenge test is an experimental test that can be carried out in a suitably accredited test laboratory according to ISO/IEC 17025:2017;

(c) It is highly advisable that the test method of the challenge test is also accredited by a third party that is in turn authorized to do so (for Italy, all certification activities are concentrated in ACCREDIA);

(d) It is the responsibility of the FBO to determine if it is necessary to schedule a challenge test for a given microorganism in a given food;

(e) It is up to the FBO to plan and set up the desired challenge test correctly, and better if technically competent personnel support the test. Strictly speaking, in fact, it is the FBO that asks the accredited analysis laboratory to conduct a challenge test in the food that the FBO has identified, with the chosen bacterium and in the manner that the FBO has established, based on the guidelines in force;

(f) It is the task of the analysis laboratory to carry out the challenge test as closely as possible to the available guidelines, using accredited methods (ISO or equivalent methods) to establish the growth potential of the bacterium subject to the challenge test;

(g) It is also the task of the analysis laboratory to carry out the experimental tests in especially dedicated environments and using bacterial strains with specific characteristics;

(h) It is the task of the analysis laboratory to draw up, at the end of the challenge test, an appropriate final report in which all the points capable of satisfying the requirements listed in the reference document used for conducting the challenge test are correctly exposed, in addition to the results obtained;

(i) It is the responsibility of the FBO who commissioned the challenge test to evaluate the report produced by the laboratory and to correctly interpret the results obtained from the challenge test;

(j) It is also the responsibility of the FBO to apply the results obtained from the challenge test to its products and production processes, adjusting them accordingly. For example, the FBO can be induced by the results of the challenge test to reduce the shelf life of its food to prevent it from becoming a potential source of an outbreak of foodborne illness;

(k) It is highly appropriate that the challenge test be conducted using different strains of the microorganism to be inoculated. In the case of foodborne bacterial pathogens, it is highly advisable to conduct the challenge test by inoculating the pathogenic bacterium in the food, in order to obtain results that are really predictive;

(l) If the challenge test concerns *L. monocytogenes*, the FBO and the laboratory will have to refer to the ISO 20976-1: 2019 standard, or in association with this, to the EURL Lm of July 2021, version 4, or create an internal reference document (integrated procedure) that meets both requirements;

(m) If the challenge test involves inoculating any bacterium other than *L. monocytogenes* into a food or a feed, the FBO and the analysis laboratory must refer only to the ISO 20976-1: 2019 standard;

(n) The challenge tests that we have commented on in this article are aimed only at determining the growth potential or maximum growth rate of bacteria and yeasts in food and feed;

(o) At the time of writing this article, in fact, a specific reference document is not yet available to design and conduct challenge tests to evaluate the effectiveness of an inactivation treatment applied to a food, against certain bacteria or yeasts (e.g., the effectiveness of a maturation process of a fermented raw salami against strains of *Escherichia coli* STEC). For this further challenge test mode, it will be necessary to wait for the publication of a new ISO standard, which is not available at the moment.

## 9. Conclusions

In this article, we have examined, in a comparative way, the texts published by ANSES and ISO that provide information on the design and conduct of challenge tests for *L. monocytogenes* or other microorganisms in food and feed. From this examination, we have obtained a series of indications that may be useful for those who want to develop and carry out challenge tests for food microbial agents.

In conclusion, we note that over the years, there has been a clear and significant evolution in the contents of the abovementioned documents, an evolution that in our opinion is not only of a technical nature. It is also an important “cultural evolution”. Apparently, the succession of ANSES guidelines and then the intervention of the specific ISO standard led to a fragmentation of the rules, with the risk of complicating the overall picture of the challenge test and creating contradictions in the rules for setting up challenge tests.

In reality, however, the progressive development of the guidelines has led to the present scenario, in which the general reference point is the ISO 20976-1 standard for the bacteria and yeasts of which the potential growth is to be determined in any food or feed. However, if the challenge test concerns only *L. monocytogenes*, the guidelines contained in the EURL Lm of 1 July 2021, which have been expressly written to integrate ISO 20976-1, must be associated constructively with the ISO standard.

It is therefore an excellent example of the integration of technical rules that may seem difficult to implement in their entirety and therefore not very user-friendly. In fact, in accordance with our personal experience, we can conclude that the rules contained in ISO 20976-1:2019 and, respectively, in EURL Lm 2021, when used in association with each other, are currently the most effective strategy for designing and conducting challenge tests that are really effective in determining the growth potential of *L. monocytogenes* and other food-related microorganisms.

The final aim of a challenge test is to be able to establish the growth potential of the experimentally inoculated bacterium in the food matrix with the maximum possible predictivity. In fact, we must not forget that the results obtained in a challenge test are not ends in themselves. Each FBO must evaluate the results of each single challenge test and interpret them in the most correct way. Only in this way can the FBO estimate in the most exact way possible what happens in the objective reality of the facts when, occasionally, foodborne pathogens such as *L. monocytogenes* contaminate a food with the risk of an outbreak of foodborne disease.

The judgment of an FBO must be based on the results of a correct challenge test to evaluate the operating conditions of its production process and, finally, the correct definition of the shelf life of its food. Ultimately, therefore, all the safety of food depends on the correct setting and implementation of a challenge test. In fact, obtaining results that are “realistic” allows FBOs to adopt the most effective strategies within their production processes and then for the subsequent storage of food during the marketing phase. On the contrary, if an FBO bases its assessments and corrective strategies on results that are at least “inaccurate” because they were obtained from an incorrectly designed challenge test, there could be serious consequences for the health of consumers.

## Figures and Tables

**Table 1 foods-11-00032-t001:** Number confirmed cases, hospitalized cases and number of death of Listeriosis oubreaks in EU/EEA (31 States) from 2007 to 2020.

Year	Number of Confirmed Cases ^1^	Hospitalized Cases	Number of Deaths
2007	1634	N.R.	165
2008	1459	N.R.	136
2009	1706	314	132
2010	1686	634	183
2011	1539	568	136
2012	1754	652	202
2013	1905	731	184
2014	2250	839	215
2015	2201	982	272
2016	2519	980	248
2017	2497	1010	232
2018	2570	1073	229
2019	2652	1260	306
2020	1931	817	168

^1^ Confirmed cases: number of sick persons clinically compatible with a diagnosis of *L. monocytogenes*; N.R.: data not reported; Source: European CDC data Base [18]. Data taken from the database and completely redrawn by the authors.

**Table 2 foods-11-00032-t002:** Some foodborne disease outbreaks of listeriosis registered around the world caused by the consumption of ready-to-eat (RTE) Foods.

Year	State	RTE Food	Confirmed Cases (*)	Deaths	Prevalent Serovar	Reference
1983	Massachusetts	Pasteurized whole milk	49	14	4b	[27]
1985	California	Soft cheese	142	48	4b	[28]
1989	Connecticut	Shrimp	10	0	4b	[29]
June–October 1993	France	Rillettes (pork meat product)	31	0	4b	[30]
August 1998	USA (multistate)	Hot dogs and deli meat	38	3	4b	[31]
2000	USA (multistate)	Deli turkey meat	29	4	Un.	[32]
November 2008	Canada	Delicatessen meat	57	24	Un.	[33]
2009	Austria and Germany	Quargel (sour milk curd cheese)	14	5	1/2a	[34]
April–July 2011	Switzerland	Cooked ham	6	0	1/2a	[35]
2009–2012	Portugal	Queijo fresco (fresh cheese)	30	11	4b	[36]
August 2012	Spagna	Latin-style fresh cheese	2	0	1/2b	[37]
2012	USA (multistate)	Italian-style cheese (ricotta salata)	22	4	1/2a	[38]
January–February 2016	Switzerland	Meat pâté	5	0	4b	[25]
May 2016	Italy	Beef ham	217 (**)	0	1/2b	[39]
January 2017–July 2018	South Africa	Processed meat products	1060	216	Un.	[40]
August 2017	Denmark and France	Probable cold-smoked salmon	5	1	Un.	[41]
December 2018	Austria	Smoked meat and liver pâté	13	0	4b	[42]

(*): Confirmed cases: number of sick persons clinically compatible with a diagnosis of *L. monocytogenes*. (**): Probable cases. They have the typical clinical features of the illness but without laboratory confirmation. Un.: undefined.

**Table 3 foods-11-00032-t003:** Key points that the FBO must keep in mind to design and implement an effective challenge test: questions and answers.

Question	Answer (According to the Guidelines Considered) [17]
What purpose can the challenge have?	A challenge test can be designed for any kind of bacterium, mold, or yeast which does not form a mycelium (adopt ISO 20976-1)
In which matrices can a challenge test be planned?	A challenge test can be carried out by inoculating microorganisms in any type of food and feed
What guidelines should we consider when planning a challenge test with *Listeria monocytogenes*?	If the challenge test concerns *Listeria monocytogenes*, in addition to the ISO 20876-1 standard, the indications provided by the ANSES EURL Lm 2021 guidelines must also be considered
What product conditions must be evaluated before setting up the challenge test?	Determine if there is a significant inter-batch difference based on the pH and A_w_ values of the individual product
How many batches of food do we need to perform a challenge test?	If the inter-batch difference is not significant, the challenge test can be performed on a single batch of food. If the inter-batch difference of the product is significant, plan and perform the challenge test on three different batches of food
How many strains of the same microorganism do we need to inoculate?	Inoculate the food with a mixture of at least two strains of the same microorganism to be tested. It would be appropriate that the inoculated strains are “wild” (strains isolated from foods of the same type as the one subject to the challenge test)
What characteristics should the individual inoculated microbial strains have?	Document as much as possible the biochemical, serological, and genomic characteristics of the wild strains that are inoculated in the food subject to the challenge test
How many “test units” must be inoculated for each scheduled analysis time?	If the purpose of the challenge test is to calculate the growth potential of the microorganism studied, inoculate a series of test units sufficient to perform analytical determinations on 3 test units at five different times of analysis, between the first day of production and the end of the product’s shelf life
How many “test units” must be inoculated for each scheduled analysis time?	If the purpose of the challenge test is to calculate the “maximum growth rate” of the microorganism studied, inoculate a series of test units sufficient to perform analytical determinations on 3 test units at eight different times of analysis, between the first day of production and the end of the product’s shelf life
Should other “test units” be programmed in addition to those with the microorganisms subjected to challenge tests?	For every three inoculated test units add a fourth “accessory unit” on which to determine the pH and A_w_ values of the food and the background microbial flora
Under what storage conditions should test units and accessory ones be kept?	Plan to keep the inoculated test units and accessory units in programmed environmental conditions, evaluating the possibility of conducting the challenge not only at refrigeration temperature, but also in conditions of “planned thermal abuse”
How should the growth potential of the inoculated organisms be calculated?	The growth potential is calculated by subtracting the logarithm of the initial concentration inoculated from the logarithm of the maximum concentration reached by the bacterium in the challenge test, regardless of the moment of analysis in which this value was reached
Are there specific concentrations to be inoculated for each individual microorganism selected?	In each single test unit inoculate a concentration of the chosen microorganism between 50 and 10,000 cfu/g (only for *Listeria monocytogenes*, inoculate in each single test unit a concentration between 50 and 200 cfu/g)
How should we document the results obtained from the challenge test?	At the end of the test, it is mandatory to draw up a final report containingall the information required by the ISO 20976-1: 2019 standard
If the challenge test is conducted on three batches, what is the growth potential that we have to consider valid?	The growth potential of the microorganism subject to challenge testing in three batches is the highest value recorded in the three lots under examination
Who is responsible for interpreting and evaluating the results obtained with a challenge test?	Based on the results obtained, it is up to the FBO to establish whether its product is a food that allows or does not allow the growth of the microorganism subject to the challenge test

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
