# Peer review of "Challenge Test as Special Tool to Estimate the Dynamic of Listeria monocytogenes and Other Foodborne Pathogens"

_foods, 2021, doi:10.3390/foods11010032_

Round 1

Reviewer 1 Report

This manuscript aims to compare the guidelines currently existing in Europe and in the world (three official documents), to obtain information that can be useful for those who have to design and implement challenge tests on food and feed. The authors have carefully compared the texts, principles or descriptions (terms or definitions) published by ANSES and ISO that provide information of using challenge tests for L. monocytogenes or other microorganisms in food issues. In general, this manuscript is well written in most parts and the results obtained from the comparison could be useful for those who want to develop or carry out challenge tests for food microbial agents.

Minor parts:

Line 186-188

It seems that several sentences here should be combined together.

Author Response

REVIEWER 1 – Minor parts:

Request 1.: Line 186-188 “It seems that several sentences here should be combined together”.

Corrective action done: We have grouped the sentences as requested by the Reviewer 1.

Reviewer 2 Report

The presented review comprehensively shows the challenges related to foodborne illness matters.

The work is written well, and I have only a few minor comments:

1. All abbreviations are not explained, e.g. GMP, GPP (line 44) or
HACCP (line 134).
2. What Articles 3 and 4 did the authors mean (lines 169 and 170)?

Author Response

REVIEWER 2 – a few minor comments:

Request 1. All abbreviations are not explained, e.g. GMP, GPP (line 44) or HACCP (line 134).

Corrective action done: we have included the explanation of the abbreviations indicated by the reviewer directly in the text of the article.

Request 2. What Articles 3 and 4 did the authors mean (lines 169 and 170)?

Corrective action done: We have included in our text the two articles of law of Regulation 852/2004 with our related comments.

SEE ALSO FILE ATTACHED

Reviewer 3 Report

I think it is useful as a review to improve the feasibility of implementing the challenge test from the user's perspective.

I think it would help the reader's understanding if it were stated more clearly which part of the ISO standard is difficult , and figures and if tables were used to promote understanding.

Although the challenge test itself is generally concerned with pathogenic microorganisms in foods, this review appears to focus on Listeria.

In the introduction, it would be better to mention about the background of food poisoning caused by RTE foods, causative pathogens, and the Codex standard, and then summarize the scope of application of the challenge test and the status of guidelines, and finally focus on challenge test about listeria in the review.

The authors should describe as a background information, the differences between standards for the number of bacteria in food in the US and EU, there is no unified standard for Listeria in food, zero tolerance in the United States and 100 cfu/g in the EU depending on food. So, why EU needs to conduct the challenge test about Listeria?

Overall, the structure of the paper seems to be incomplete, so it is needed to restructure and consider resubmitting.

  1. The title does not reflect the content.It would be better to have a title that shows the comparison and the suggestion of a better way for challenge test in practical use. Also, the title and content should focus more on the Listeria challenge test. 
  2. I think the conclusion is that it seems better to combine ISO and ANSES EURL Lm 2021 for better challenge testing, but it would be easier to understand if the flow diagram or tables were used to show where the essence of ANSES was added based on ISO.
  3. Regarding the comparison between ANSES EURL and  ISO, it would be good to summarize the differences in a table. It would help the reader to understand. 
  4. paragraph 5 has only session 5.1 'Usefulterms and definitions', where is 5.2?
  5. Abbreviations are not explained enough, such as EFSA, GMP, GPP,.
  6. Bacterial species are not italicized.
  7. ISO20876-1 was in the L286, but others are ISO20976-1.

Author Response

REVIEWER 3 – many comments and suggestions:

Request 1.: I think it would help the reader's understanding if it were stated more clearly which part of the ISO standard is difficult, and figures and if tables were used to promote understanding.

Corrective action done: we have inserted three tables in the text of the article that contribute to making our text more understandable, in particular Table 3 aims to make the various indications contained in the two main guidelines assessed more applicable by the FBO (the ISO 20976-1 standard and the ANSES EURL Lm 2021).

Request 2. "Although the challenge test itself is generally concerned with pathogenic microorganisms in foods, this review appears to focus on Listeria."

Corrective action done: we changed the title of the article by focusing attention on Listeria monocytogenes. On the other hand, only ISO 20976-1 takes into consideration all bacteria, yeasts and molds, the ANSES guidelines have always and only focused on L. monocytogenes, so it is almost mandatory that in the text we deal essentially by L. monocytogenes.

Request 3. "In the introduction, it would be better to mention about the background of food poisoning caused by RTE foods, causative pathogens, and the Codex standard, and then summarize the scope of application of the challenge test and the status of guidelines, and finally focus on challenge test about listeria in the review."

Corrective action done: we have added to our writing a special chapter (chapter 3) in which we have framed the epidemiological aspects that justify the planning of challenge tests in RTE foods.

Request 4. "The authors should describe as a background information, the differences between standards for the number of bacteria in food in the US and EU, there is no unified standard for Listeria in food, zero tolerance in the United States and 100 cfu / g in the EU depending on food. So, why EU needs to conduct the challenge test about Listeria? "

Corrective action done: we have added a special chapter to our paper (chapter 3) in which we have illustrated what are currently the microbiological compliance criteria for L. monocytogenes in RTE foods.

Request 5. "Overall, the structure of the paper seems to be incomplete, so it is needed to restructure and consider resubmitting-"

Corrective action done: with our various additions to the initial article, we believe we have integrated the article with additional, useful content.

Request 6. "The title does not reflect the content. It would be better to have a title that shows the comparison and the suggestion of a better way for challenge test in practical use. Also, the title and content should focus more on the Listeria challenge test. "

Corrective action done: we have changed the title of the article.

Request 7. "I think the conclusion is that it seems better to combine ISO and ANSES EURL Lm 2021 for better challenge testing, but it would be easier to understand if the flow diagram or tables were used to show where the essence of ANSES was added based on ISO"

Corrective action done: we have added Table 3 to the article which summarizes in summary form all the useful information that an FBO must know to plan a challenge test based on the ISO 20976-1 standard and together on the ANSES EURL Lm 2021.

Request 8-9. "I think the conclusion is that it seems better to combine ISO and ANSES EURL Lm 2021 for better challenge testing, but it would be easier to understand if the flow diagram or tables were used to show where the essence of ANSES was added based on ISO "Regarding the comparison between ANSES EURL and ISO, it would be good to summarize the differences in a table. It would help the reader to understand ".

Corrective action done: we have added Table 3 to the article which summarizes in summary form all the useful parts that an FBO must know to plan a challenge test based on the ISO 20976-1 standard and together on the ANSES EURL Lm 2021.

Request 10. "paragraph 5 has only session 5.1 'Usefulterms and definitions', where is 5.2?".

Corrective action done: this is a simple typo in the text, we have not seen a point 5.2. for this chapter and therefore we have deleted all the numbers entered ..

Request 11. "Abbreviations are not explained enough, such as EFSA, GMP, GPP ,.".

Corrective action done: we have included the explanation of the abbreviations indicated directly in the text of the article, with their meanings.

Request 12. "Bacterial species are not italicized.".

Corrective action done: We have italicized the bacterial species as required.

Request 13. "ISO20876-1 was in the L286, but others are ISO20976-1.".

Corrective action done: we have corrected the typo, as requested.

see also file attached

Round 2

Reviewer 3 Report

Thank you for resubmitting the paper. I think this will be a very clear and useful review or guide for food producers who need to implement challenge test in the future.
In particular, the background information on the outbreaks of listeria food poisoning caused by RTE foods and the description of the current regulations for listeria in foods among countries are well organized and will help the reader to understand the situation.

Please correct the notation for listeria, as it is not italicized.

L224, 294, 318, 590, 962

Other than that, I have no other comments.

I hope this will be a useful review.

Author Response

Following our review, Reviewer 3 reported some minor revisions to do.

Request 1.: Please correct the notation for listeria, as it is not italicized.

L224, 294, 318, 590, 962

Response 1: I reviewed the full text of the article and proceeded to italicize all the species names “L. monocytogenes”, including those mentioned in the references.